# The neural transdifferentiation potential of bone marrow mesenchymal stem cells

Marie Ytterdal[1,2*], Casper Eugen Sandvik[2], Trygve Holmøy[3,4], Lars Bø[1,2], Christopher Elnan Kvistad[2], Torbjørn Kråkenes[1,2]

1 Department of Clinical Medicine, Faculty of Medicine, University of Bergen, Bergen, Norway, 2 Neuro-SysMed, Department of Neurology, Haukeland University Hospital, Bergen, Norway, 3 Department of Neurology, Akershus University Hospital, Lørenskog, Norway, 4 Institute of Clinical Medicine, University of Oslo, Oslo, Norway

* marie.ytterdal@helse-bergen.no

## Abstract

### Background

Multiple sclerosis is a chronic inflammatory disease of the central nervous system characterized by demyelination and neuronal degeneration. Currently, there is no treatment that can promote remyelination or axonal repair and thus improve patient outcomes. Mesenchymal stem cells (MSCs) have emerged as a promising therapeutic approach in multiple sclerosis due to their regenerative and immunomodulatory properties. Priming MSCs toward neural- or glial-like cells may enhance their therapeutic potential.

### Methods

Bone marrow MSCs from multiple sclerosis patients were cultured in differentiation medium containing growth factors for induction of glial and neural transdifferentiating process. Functional assays were used to characterize the proliferation and migration rates of undifferentiated and transdifferentiating MSCs. In addition, immunostaining with five different markers was performed as well as mass spectrometry to investigate and compare protein expressions among the cell types.

### Results

Proliferation and migration rates of MSCs decreased in glial and neural transdifferentiating cells. The immunostaining showed expression of neuro-glial markers, and the mass spectrometry data demonstrated substantial variance in protein expression in transdifferentiating MSCs compared to undifferentiated MSCs. Glial and neural markers were upregulated in the transdifferentiating MSCs, while MSC markers were downregulated.

**Data availability statement:** All relevant data are within the paper and its Supporting Information files.

**Funding:** M.Y.: PhD stipend from Helse Vest, Bergen, Norway. The funders had no role in study design, data collection and analysis, decision to publish, or preparation of the manuscript.

**Competing interests:** The authors have declared that no competing interests exist.

## Conclusion

The results demonstrate that MSCs can be guided toward a neuro-glial lineage in vitro and that the transdifferentiating process continues over several weeks. Future research should investigate the in vitro and in vivo therapeutic differences between different transdifferentiating lengths.

## Introduction

Multiple sclerosis (MS) is a chronic inflammatory disease of the central nervous system (CNS), characterized by inflammation, demyelination, gliosis and neuronal loss ultimately leading to permanent disability [1]. Although spontaneous remyelination can occur, it is often incomplete due to reduced maturation of oligodendrocyte precursor cells (OPCs) into myelinating oligodendrocytes. Currently there are no approved treatments that effectively promote remyelination [2].

The most common clinical form is relapsing-remitting MS (RRMS), which accounts for approximately 85% of initial diagnoses [3]. Over time, a significant proportion of RRMS patients develop secondary progressive MS, characterized by progressive neurological decline. A smaller group (5–15%) presents with primary progressive MS from onset [4,5]. MS typically debuts in adults between 20–40 years old and is the leading cause of non-traumatic neurological disability in this population [6]. Although disease-modifying therapies can reduce disease activity in RRMS, their efficacy in progressive MS is limited [5]. As such, there is a critical need for innovative treatments that support remyelination and axonal repair, with the potential to significantly improve patients' quality of life.

Mesenchymal stem cells (MSCs) have emerged as promising candidates for regenerative therapies in MS. These multipotent stem cells can self-renew and differentiate into various mesodermal cell types, including osteoblasts, chondrocytes, and adipocytes [7]. Importantly, MSCs can be readily isolated and expanded from multiple sources, such as bone marrow (BM), adipose tissue, umbilical cord blood and placenta. Beyond their differentiation potential, MSCs have immunomodulatory, anti-inflammatory, and neuroprotective properties, highlighting their potential for treating neurodegenerative and autoimmune diseases [8,9].

Upon exposure to inflammatory stimuli, MSCs can modulate immune responses by secreting cytokines and growth factors that promote angiogenesis, neurogenesis, extracellular matrix remodeling, and the recruitment or differentiation of tissue-resident progenitor cells [10]. Compared to other stem cell therapies, such as induced pluripotent stem cells (iPSCs), embryonic stem cells, and neural stem cells (NSCs), MSCs represent no major ethical concern, are more accessible and carry a lower risk of tumorigenicity [11,12]. Furthermore, their ability to home to areas of injury and influence the CNS microenvironment has led to numerous clinical trials investigating MSC therapy in MS and other inflammatory conditions [13,14].

Several clinical trials have demonstrated that MSC administration, both intrathecally and intravenously, is generally safe and well tolerated in MS patients [15–17].

However, results regarding clinical efficacy are varied. In a clinical trial treating RRMS patients with intravenously BM-MSCs no significant benefits were reported [15]. A limitation of this study is the intravenous administration of MSCs in patients with RRMS, a group in which highly effective therapies are already available, and therefore limiting the need for MSC therapy in this population [15]. Llufriu et al. included 9 patients with RRMS and administrated a single dose of MSCs intravenously in a placebo cross-over design. They observed reductions in inflammatory MRI activity, suggesting anti-inflammatory effects after MSC treatment [16]. However, limitations of this study include the small sample size and lack of statistical significance [16]. Study by Petrou et al. [17] included 48 patients with progressive MS in a randomized controlled trial comparing intrathecal and intravenous administrations and single versus repeated doses. Results showed that intrathecal administration gave greater benefits than intravenous delivery. The intrathecal delivery of MSCs demonstrated improved clinical scores in active progressive MS patients and concluded that MSC transplantation is safe in progressive MS patients [17].

Interestingly, MSCs may possess the capacity to transdifferentiate into ectodermal cell types, including neural progenitor cells and oligodendrocyte progenitor cells (OPCs), potentially enhancing their reparative potential within the CNS [18]. Transdifferentiation refers to the ability of cells to develop into cells outside their original germ layer lineage [19]. In the context of MS, transdifferentiation of MSCs into neural or oligodendroglial lineages prior to transplantation may increase the production of neurotrophic and glial growth factors, thereby enhancing their neuroprotective effects. In this study, we use the term transdifferentiating MSCs to describe MSCs that are undergoing molecular and phenotypic changes toward neural or glial lineages under defined in vitro conditions. This term is used to describe the shift toward neural- or glial-associated progenitor states rather than terminal differentiation into mature neuronal or glial cell types.

Transdifferentiation of MSCs typically involves modifying the culture environment using biological or chemical stimuli, most notably growth factors such as epidermal growth factor (EGF), basic fibroblast growth factor (bFGF), hepatocyte growth factor (HGF), and platelet-derived growth factor-AA (PDGF-AA), to direct MSC differentiation toward neuro-glial lineages [20,21]. In vitro studies have demonstrated that MSCs cultured in NSC medium adopt a neuron-like morphology, characterized by elongated processes, and expression of neuro-glial markers such as neuronal nuclei (NeuN), glial fibrillary acidic protein (GFAP) and neuron-specific enolase [22–24]. Similarly, exposure to bFGF and PDGF-AA has been shown to induce expression of oligodendrocyte markers, including oligodendrocyte transcription factor 1 (Olig1) and myelin basic protein (MBP), suggesting the potential for OPC-like differentiation [25,26]. However, most studies have relied only on morphological changes and/or analyses such as PCR, flow cytometry, and western blotting of selected makers. Furthermore, most studies have not characterized the transdifferentiation process over time.

Due to the limited characterization of the transdifferentiation process, some studies question the extent of MSC plasticity and their ability to truly transdifferentiate into functional neural lineages [27]. In the present study, we aimed to perform a comprehensive deep molecular characterization of the changes occurring during and after transdifferentiating MSCs in glial and neural directions, including morphology, immunostaining, cell growth and migration assays, as well as mass spectrometry analysis from several time points.

## Materials and methods

### Source and characterization of MSCs

Human bone marrow-derived MSCs were obtained from the SMART-MS biobank (Department of Neurology, HUS) which was approved by the ethical board (REK#159326). All patients provided written consent to participate in the study. The SMART-MS study registered at ClinicalTrials.gov (NCT04749667) is a phase I/II, double-blinded, placebo controlled clinical trial. The trial was completed in 2025 and enrolled 18 patients with progressive MS that were treated with autologous MSCs intrathecally. For the present study, MSCs from seven patients were randomly selected for further analysis (Table 1).

MSCs were isolated and expanded at the Institute for Transfusion Medicine in Ulm, Germany. BM aspirates were collected from the iliac crest under local anesthesia, anticoagulated with heparin, and transported to Ulm, Germany. MSCs

**Table 1. MSC donor demographics.**

| Donor | MS Subtype | Gender | Age | Disease Duration (years since the diagnosis) | EDSS |
|-------|-----------|--------|-----|----------------------------------------------|------|
| D1 | Secondary progressive | Male | 46 | 8.4 | 7 |
| D2 | Secondary progressive | Female | 41 | 7.9 | 4 |
| D3 | Primary progressive | Male | 52 | 13.3 | 6 |
| D4 | Secondary progressive | Male | 47 | 8.4 | 6 |
| D5 | Secondary progressive | Female | 54 | 15 | 6.5 |
| D6 | Secondary progressive | Male | 50 | 5 | 6 |
| D7 | Primary progressive | Female | 48 | 9.5 | 7 |

were isolated and expanded under GMP conditions. All MSCs fulfilled the *minimal criteria for defining multipotent mesenchymal stromal cells* as established by the International Society for Cellular Therapy [28]. This included adherence to plastic under standard culture conditions, expression of surface markers CD105, CD90 and CD73, and lack expression of CD45, CD34, CD14 or CD11b, CD79α or CD19, and HLA-DR.

## Cultivation of MSCs

Culture flasks were coated with Cellstart substrate (Gibco, #A10142-01) diluted 1:500 in PBS with calcium and magnesium (PBS⁺; Gibco, #14040–091) and incubated at 37°C for 60 minutes. StemPro MSC medium kit which includes basal medium and XenoFree Supplement (serum- and xeno-free; Gibco, #A1067501) supplemented with 2 mM GlutaMAX-I (1% v/v; Gibco, #A12860-01) was used for growth and expansion of the MSCs. All cells were incubated at 37°C degrees and 5% $CO_2$. Cells were passaged using enzymatic detachment with trypsin. Passages 1–6 were used for the experiments.

## Transdifferentiating MSCs

StemPro NSC medium kit with basal medium, growth supplement and 20 ng/mL EGF, 20 ng/mL bFGF (serum-free kit; Gibco, #A10509-01) and 2 mM GlutaMAX-I (1% v/v) was used to induce transdifferentiating MSCs toward a neural direction (N-MSC). Transdifferentiating toward a glial direction (G-MSC), EGF was exchanged with 20 ng/mL platelet-derived growth factor AA (PDGF-AA; Fisher Scientific, #PHG0035). Cells were not passaged during the process. Medium was changed every other day.

## Morphological assessment

The morphology of the different cell types was assessed using a Leica Mateo brightfield microscope using a 10x objective. All seven donors were evaluated both before and after the transdifferentiating process.

## Cell migration assay

Migration capacity of MSCs and transdifferentiating MSCs was compared using a wound healing assay. For this assay three different patients MSCs were used. Approximately 40.000 MSCs were seeded per well in a 96-well ImageLock microplate (Sartorius, #BA-04855) in StemPro MSC medium and incubated overnight. The following day, a scratch ~700–800 µm wide was created in the center of each well using a WoundMaker (Sartorius). Wells were rinsed twice with PBS⁺ to remove detached cells and subsequently incubated with one of the following media: StemPro MSC, StemPro glial, or StemPro neural. Plates were incubated in Sartorius IncuCyte S3 Live-Cell Analysis System (available through the Molecular Imaging Center at University of Bergen), and images were captured every hour for 48 hours. Migration was analyzed using the Sartorius IncuCyte analysis software.

## Cell proliferation assay

To compare the proliferation rate of MSCs in different culture media, 10.000 MSCs per well were seeded in a 96-well ImageLock microplate and cultured in StemPro MSC, StemPro glial or StemPro neural medium. The experiment was done three times using cells from three different donors. The IncuCyte System scanned live cells every second hour for three days. The resulting timelapses were analyzed in the Sartorius IncuCyte software.

## Statistics of migration and proliferation assay

Migration and proliferation data were analysed using linear mixed-effects models to account for repeated measurments over time. Experimental group, time, and their interaction were included as fixed effects, and experiment was included as a random intercept to account for variability between independent experiments. Models were fitted using maximum likelihood estimation, and the significance of interaction terms was assessed using Wald test.

Proliferation data were modelled with time as a continuous variable, whereas migration data were modelled with log-transformed time to capture non-linear migration dynamics. Model-based estimated marginal means at 72 hours were obtained using post hoc contrasts. Statistical analyses were performed in Stata. A p-value $< 0.05$ was considered statistically significant.

## Antibody staining

MSCs were seeded onto poly-D-lysine coated coverslips (Fisher Scientific, #NC0343705) and cultured in StemPro MSC medium until 80% confluency was reached. Transdifferentiating MSCs was induced by replacing the culture media with either StemPro glial or StemPro neural medium and cells were cultured for three weeks.

Transdifferentiating MSCs were fixed and stained for glial and neuronal markers after three weeks in culture. Cells were fixed in 4% paraformaldehyde for 10 minutes at room temperature and permeabilized for 5 minutes with 0.5% Triton X-100 in PBS$^+$. Next, a 30% fish serum solution (Thermo Scientific, #37527) was added and incubated for 30 minutes. Primary antibodies were diluted 1:100 in 10% fish serum and incubated for 1 hour at room temperature. The following antibodies were used: anti- βIII tubulin (Abcam Cat# ab18207, RRID:AB_444319), anti-SOX2 (Abcam Cat# ab97959, RRID:AB_2341193), anti-PDGFRα/platelet-derived growth factor receptor alpha (Abcam Cat# ab96569, RRID:AB_10687154), anti-NEFL/neurofilament light polypeptide (Thermo Fisher Scientific Cat# PA1–32240, RRID:AB_2149916) and anti-GFAP (Thermo Fisher Scientific Cat# PA5–16291, RRID:AB_10980769). Secondary antibodies (Thermo Fisher, #A-21245, #A-21235) were applied at 1:500 for 1 hour in the dark. Coverslips were mounted using ProLong Glass Antifade with NucBlue Stain (Thermo Fisher, #P36981). Fluorescence images were taken using an Olympus VS120 S6 Slide scanner.

## Quantification of immunofluorescence staining

Quantitative analysis was performed from two independent transdifferentiating experiments derived from two different MSC donors. For each coverslip with cells, five randomly selected fields (1000 µm x 1000 µm) were analyzed using QuPath software (v0.6.0-x64) with the positive cell detection algorithm.

For each marker and condition, this resulted in a total of ten measurements (five fields per donor). The percentage of marker-positive cells was calculated for each field and normalized to undifferentiated MSC controls, which were set to 100%. Values for G-MSCs and N-MSCs are presented as relative percentages compared to MSC controls.

Statistical comparisons between MSCs and G-MSCs, and between MSCs and N-MSCs, were performed using Welch's t-test. Data are presented as mean values with 95% confidence intervals. Statistical significance was defined as p $< 0.05$.

## Cell sample lysis

Cell protein lysates were prepared from 80% confluent flasks of MSCs, G-MSCs and N-MSCs. Cells were washed three times with ice-cold PBS before adding lysis buffer consisting of 5% SDS (Invitrogen, #24730−020) in 0.1 M Tris-HCl (pH 7.6; Invitrogen, #15568−025). The collected viscous lysate was heated at 95°C for 7 minutes.

## Protein quantification

Protein concentration was measured using the Pierce BCA Protein Assay Kit (Thermo Fisher, #23227) according to the manufacturer's instructions. Absorbance was measured at 560 nm using a Tecan Infinite F50 microplate reader, and data were analyzed in Microsoft Excel.

## Mass spectrometry and data processing

Protein samples were digested into peptides and analyzed by The Proteomics Unit (PROBE) at the University of Bergen using liquid chromatography with tandem mass spectrometry (LC–MS/MS). Briefly, peptides were separated on a nano-Ultra-Performance Liquid Chromatography (nano-UPLC) system employing a specialized column and a solvent gradient to achieve separation based on physicochemical properties. Separated peptides were ionized and introduced into an Orbitrap Eclipse Tribrid mass spectrometer equipped with a High-Field Asymmetric Waveform Ion Mobility Spectrometry (FAIMS) Pro interface to improve ion selection. Data acquisition began with an MS1 full scan of peptide ions, followed by data-independent acquisition (DIA). In DIA mode, predefined m/z windows were sequentially isolated, fragmented by tandem MS (MS/MS), and the resulting fragment ions analyzed for peptide identification and quantification.

Raw DIA files were processed using Spectronaut (v19, Biognosys) in directDIA mode. A project-specific spectral library was generated with the integrated Pulsar search engine against the UniProtKB/Swiss-Prot human proteome (downloaded May 2024), supplemented with a common contaminants database (downloaded March 2022). Search parameters specified Trypsin/P digestion with up to two missed cleavages. Carbamidomethylation of cysteine was set as a fixed modification, while methionine oxidation and protein N-terminal acetylation were included as variable modifications. A 1% false discovery rate (Q-value < 0.01) was applied at both the precursor and protein levels. Quantification was based on MS2 fragment ion intensities, with no imputation applied during Spectronaut processing.

Protein quantity reports were further processed in Python (v3.13.2) using pandas (v2.2.3). Entries corresponding to contaminants were removed. To filter for consistently quantified proteins, any protein detected in only one of three technical replicates for a given sample was removed. Proteins lacking quantifiable abundance in all samples after this filtering step were also removed. Remaining intensity values were log2-transformed transformed using the NumPy library (v2.2.4 to stabilize variance across the dynamic range of the samples. Finally, non-biological variance across acquisition batches was corrected using the ComBat algorithm (pyComBat library v0.3.3), with batch assignment defined by mass spectrometry acquisition run.

For evaluation of transdifferentiating away from the baseline MSC state, a predefined panel of markers was assessed. MSC identity was tracked with THY1/CD90, ENG/CD105, CD44, ALCAM/CD166, and PDGFRB. Differentiation towards neural lineage was assessed using NES, MSI1, NOTCH2/JAG1, FGF2-responsive signaling components, and early neuronal program markers (NCAM1, GAP43, DPYSL3/CRMP4, SNAP25). Glial differentiation was evaluated using SLC1A3/GLAST and GLUL (astrocytic), PDGFRa (OPC), GJC2/Connexin-47, TF, and MYRF (oligodendroglial), and ICAM1. Proliferation control markers (MKI67, MCM2, MCM4, TOP2A, PCNA, CCNB1, CDK1) were also used to detect changes in proliferative state.

Functional enrichment analysis of the cell proteomic changes was performed using STRING-DB (v12.0). Ranked protein lists were uploaded in "proteins with values" mode, providing log2 fold change as the per-protein value. Enrichment was evaluated against the KEGG pathway database within STRING, using STRING's multiple-testing correction. Pathways were considered significant at FDR < 0.05. Enrichment outputs were exported, and pathway figures were generated using STRING's built-in enrichment visualization tool.

## Results

### Morphological features of MSCs and transdifferentiating MSCs

Undifferentiated MSCs maintained a spindle-shaped, fibroblast-like morphology throughout the three-week culture period and continued to proliferate, requiring multiple passages (Fig 1). In contrast, G-MSCs and N-MSCs proliferated at a much

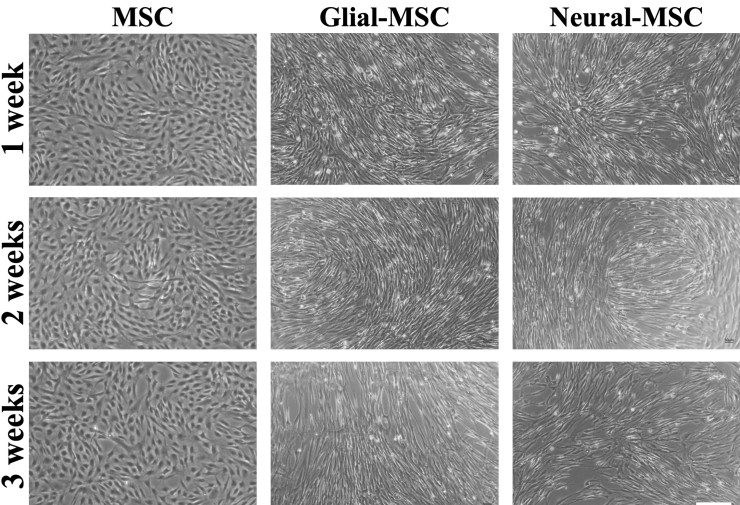

**Fig 1. Representative phase-contrast images of MSCs, G-MSCs and N-MSCs under indicated culture conditions.** Images are provided for illustrative purposes only. Scale bar = 200 µm.

slower rate and did not require splitting during the three-week transdifferentiation process. Transdifferentiating MSCs exhibited morphological changes relative to undifferentiated MSCs, including cell elongation and aligned growth patterns. These features were consistent with a neural/glial-like progenitor phenotype, while remaining less complex with stellate morphology and extensive process branching typically for neurons and oligodendrocytes [29].

### The migration rate of G-MSCs and N-MSCs are slower

Migration capacity was assessed using a scratch wound assay over a 48-hour period (Fig 2). The figure shows averaged data from three MSC donors, with experiments performed independently for each donor. Migration over time was analyzed using a linear mixed-effects model with log-transformed time. A significant group-by-time interaction was observed ($X^2 = 85.7$, $p < 0.001$), indicating that glial- and neural-like MSC cultures exhibited slower migration rates over time compared with undifferentiated MSCs. After finalizing the 48-hour experiment period, MSCs had closed the wound by 95%, while G-MSCs and N-MSCs achieved approximately 78% closure.

### The proliferation of G-MSCs and N-MSCs are slower

MSCs were cultured in either StemPro MSC, StemPro glial or StemPro neural medium, and proliferation was measured for 72 hours. MSCs proliferated significantly faster compared to G-MSCs and N-MSCs (Fig 3). The assay was done in triplicate with MSCs from three different patients. The average is shown in the graph. Proliferation over time was analysed using a linear mixed-effects model. A significant group-by-time interaction was observed ($X^2 = 415.7$, $p < 0.001$), indicating that proliferation rates differed between MSC cultures and glial/neural cultures. While no significant differences in baseline proliferation were detected ($p = 0.30$), MSC cultures exhibited a significantly steeper increase in proliferation over time. After 72 hours in culture, MSCs reached 70% confluence, while G-MSCs and N-MSCs were 38% confluent.

### Immunofluorescence staining of neural and glial markers

Immunofluorescence staining was performed to assess the presence of neural and glial lineage markers in MSCs, G-MSCs and N-MSCs (Fig 4). G-MSCs and N-MSCs showed a significant increase in NEFL and SOX2 positive cells compared with MSCs, wheras no significant differences was observed for GFAP and β-III-tubulin. PDGFRα expression was

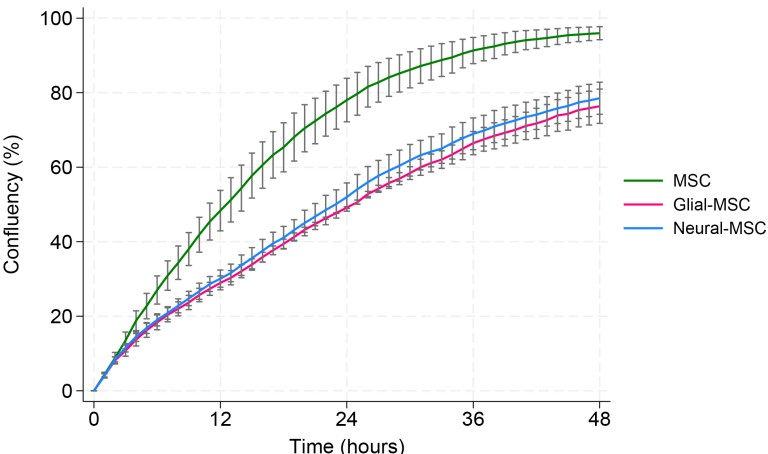

**Fig 2. Migration rate of MSCs, G-MSCs and N-MSCs.** MSCs were seeded in wells and grown until confluent before a scratch was made in the middle of each well. Three different culture media was used, StemPro, StemPro glial and StemPro neural. The confluency of the scratch wound was measured for 48 hours in Sartorius IncuCyte. The migration over time was analyzed using a linear mixed-effects model with log-transformed time. The graph shows that undifferentiated MSCs closed the scratch wound significantly faster compared to G-MSCs and N-MSCs ($X^2 = 85.7$, $p < 0.001$). The lines in the figure represent the average from three independent experiments using MSCs from three different patients. Standard errors are marked on the graph. D1, D4 and D7 were used in this experiment.

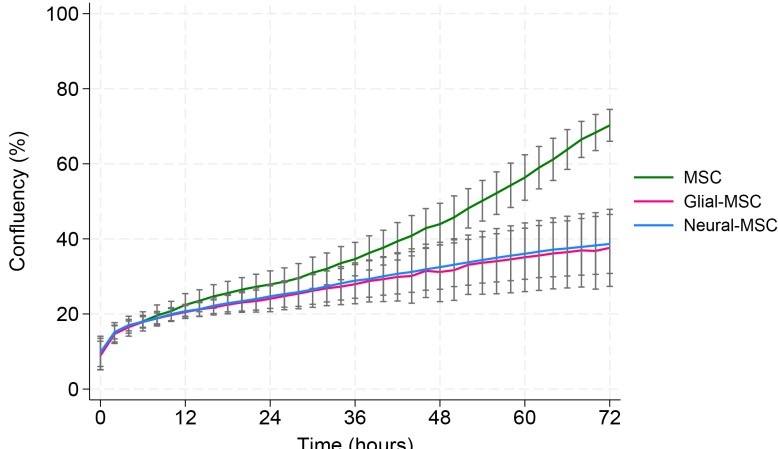

**Fig 3. Proliferation rates of MSCs and transdifferentiating MSCs.** MSCs were cultured in either StemPro, StemPro glial or StemPro neural and proliferation was measured for 72 hours. The graph shows that MSCs (green) proliferated significantly faster compared to G-MSCs (pink) and N-MSCs (blue) during the three days in culture ($p < 0.001$). The lines in the figure represent the average from three independent experiments using MSCs from three patients. Standard error bars are marked on the graph. D1, D4 and D7 were used in the experiment.

significantly increased in G-MSCs compared with MSCs, whereas no significant change was observed between MSCs and N-MSCs.

## Transdifferentiating alters MSC proteome

To assess how the duration of neural and glial differentiation influences the MSC proteome, mass spectrometry based proteomic analysis was performed after 7, 14 and 21 days of differentiation. Due to sample availability across all time

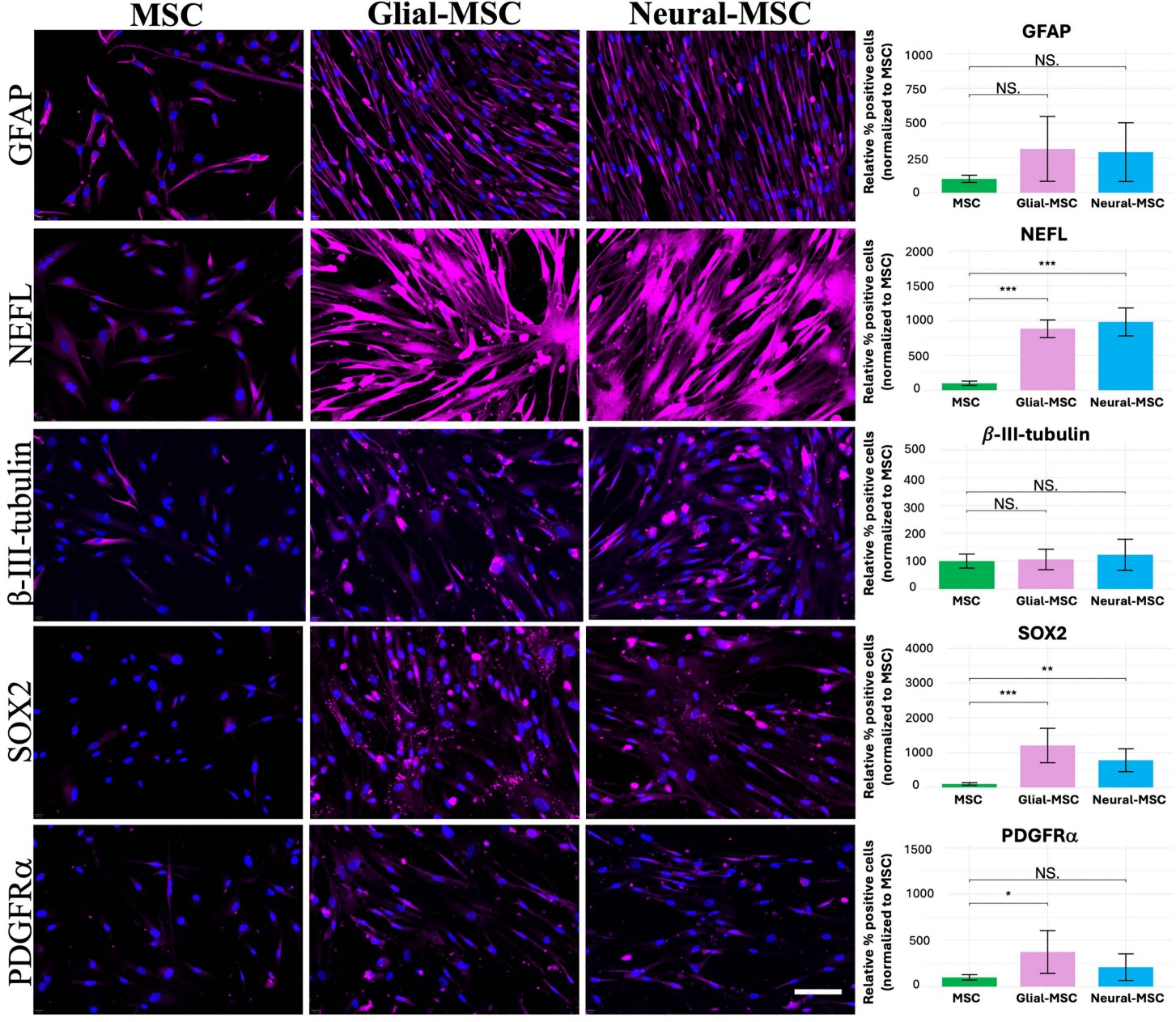

**Fig 4. Immunofluorescence staining of glial and neural markers in MSCs, G-MSCs and N-MSCs.** After three weeks of differentiation, MSCs, G-MSCs and N-MSCs were stained for GFAP, NEFL, β-III-tubulin, SOX2 and PDGFRα (magenta). Nuclei were counterstained with DAPI (blue). Scale bar = 100 µm. NEFL and SOX2 were significantly increased in both G-MSCs and N-MSCs compared to undifferentiated MSCs, while GFAP and β-III-tubulin showed no significant differences. PDGFRα were significantly increased in G-MSCs compared the MSCs, while no significant change was seen in N-MSCs. Statistical significance was assessed using the Welch's t-test. Significance levels are indicated as: * ($p < 0.05$), ** ($p < 0.01$), *** ($p < 0.001$), not significant (NS). D4 and D6 were used for these experiments.

points, Fig 5 shows a PCA plot of MSCs from a single donor. Samples collected at day 7 were analyzed in duplicate, whereas samples collected at days 14 and 21 were analyzed in triplicate. The proteome of MSCs was distinctively different from G-MSCs and N-MSCs (Fig 5). The first principal component (PC1) accounted for 42.3% of the total variance. The

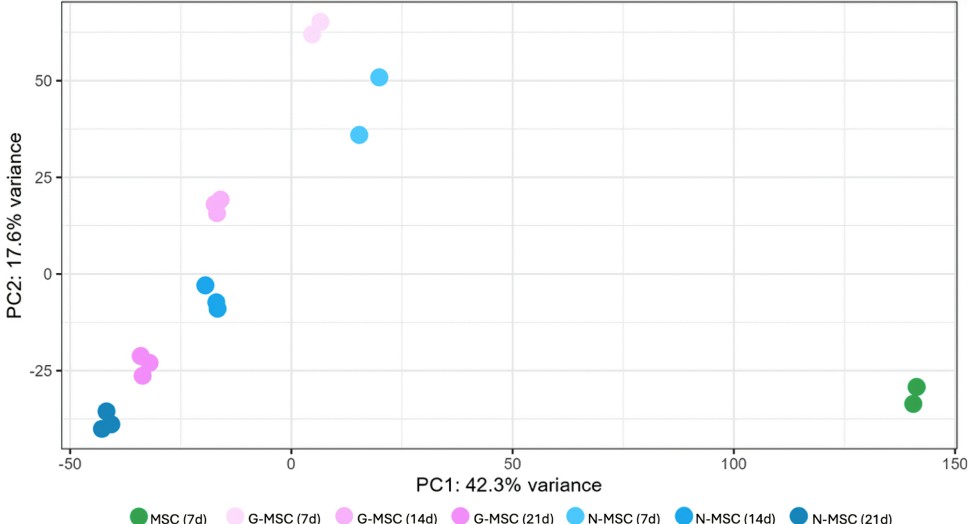

**Fig 5. PCA of mass spectrometry based proteomics from MSCs undergoing neural and glial differentiation.** Proteomic analysis was performed at 7, 14, and 21 days of differentiation using MSCs from a single donor. Due to the 18-sample capacity of a single mass spectrometry run, samples collected at 7 days were analyzed in duplicate, whereas samples collected at days 14 and 21 were analyzed in triplicate. MSCs (green) are separated from the other samples, where greater distance indicates larger proteomic differences. The transdifferentiating MSCs (pink and blue) progressed further away from the MSCs over time, demonstrating ongoing transdifferentiation during the experimental period. D7 were used in the experiment.

PCA plot revealed a clear separation of samples according to differentiation condition and time point, indicating progressive proteomic changes during differentiation. The PCA plot revealed that G-MSCs clustered together, as did N-MSCs, indicating consistent proteomic profiles within each differentiation condition.

## Proteomic overview of MSCs versus transdifferentiating MSCs

Proteomic profiling revealed distinct expression patterns among MSCs, G-MSCs and N-MSCs (Fig 6). Hierarchical clustering of seven biological replicates, six different MSC donors, showed clear separation between MSCs and transdifferentiating MSCs after 21 days in culture. The heatmap shows that transdifferentiating cells exhibited reduced expression of MSC identity markers THY1, CD44, CD105 (ENG) and ALCAM compared to undifferentiated MSCs (Fig 6). Also, a reduction of proliferation markers MKI67, MCM2 and CDK1 is shown in transdifferentiating MSCs compared to undifferentiated MSCs. The downregulation of several proliferation markers in transdifferentiating MSCs confirms the results from the proliferation assay (Fig 3). The glial lineage markers (PDGFRα, GLUL, SLC1A3) were upregulated in the G-MSCs compared to undifferentiated MSCs and N-MSCs. The neural lineage markers (NCAM1, GAP43, FGF2) were upregulated in N-MSCs compared to undifferentiated MSCs. These changes indicate a partial shift toward neuro-glial phenotypes and reduced proliferative capacity following transdifferentiating process.

## Enrichment of KEGG pathways in transdifferentiating MSCs

KEGG pathway enrichment analysis revealed distinct transcriptional programs in neural and glial MSCs compared to undifferentiated MSCs (Fig 7). The results of pathway analysis revealed several pathways that were upregulated in N-MSCs, including regulation of actin cytoskeleton, focal adhesion, PI3K-Akt signaling pathway, ECM-receptor interaction and axon guidance. Downregulated pathways in N-MSCs involved ribosome, spliceosome, RNA transport, amyotrophic lateral sclerosis (ALS), Huntington disease, and Parkinson disease.

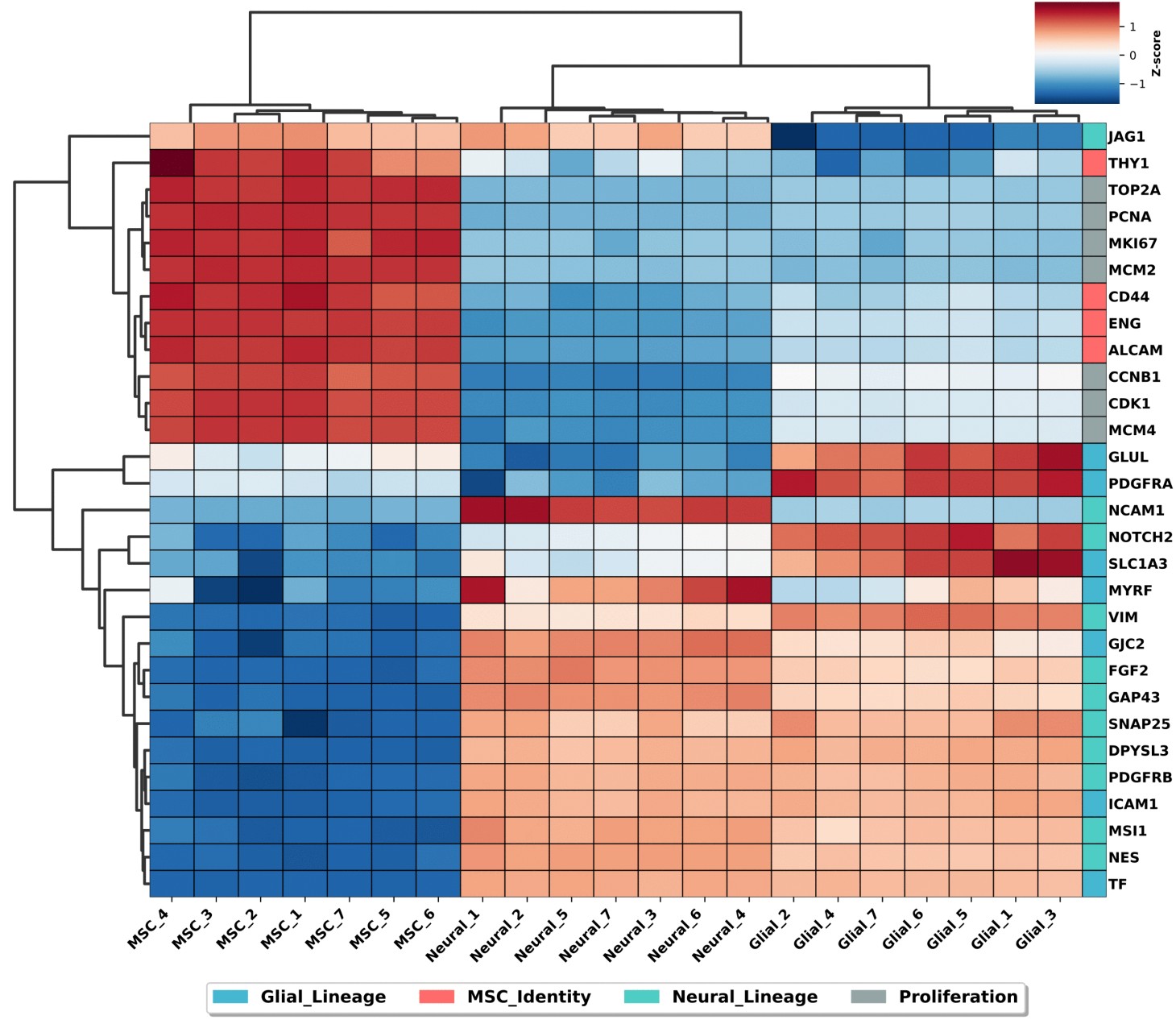

**Fig 6. Proteomic heatmap showing differential protein expression in MSCs, glial and neural MSCs.** Heatmap displays z-score normalized expression levels of selected proteins related to MSC identity (red labels), proliferation (grey), glial lineage (blue) and neural lineage (green) across 7 biological replicates. Clustering reveals downregulation of MSC and proliferation markers, and upregulation of neuro-glial markers in transdifferentiating cells, indicating a shift toward neuro-glial phenotypes. D1, D2, D3, D4, D5 and D7 were used in this experiment.

In G-MSCs, upregulated pathways included metabolic pathways, focal adhesion, regulation of actin cytoskeleton, PI3K-Akt signaling pathway, ECM-receptor interaction and axon guidance. The downregulated pathways included ribosome, spliceosome, RNA transport, RNA degradation, DNA replication and cell cycle.

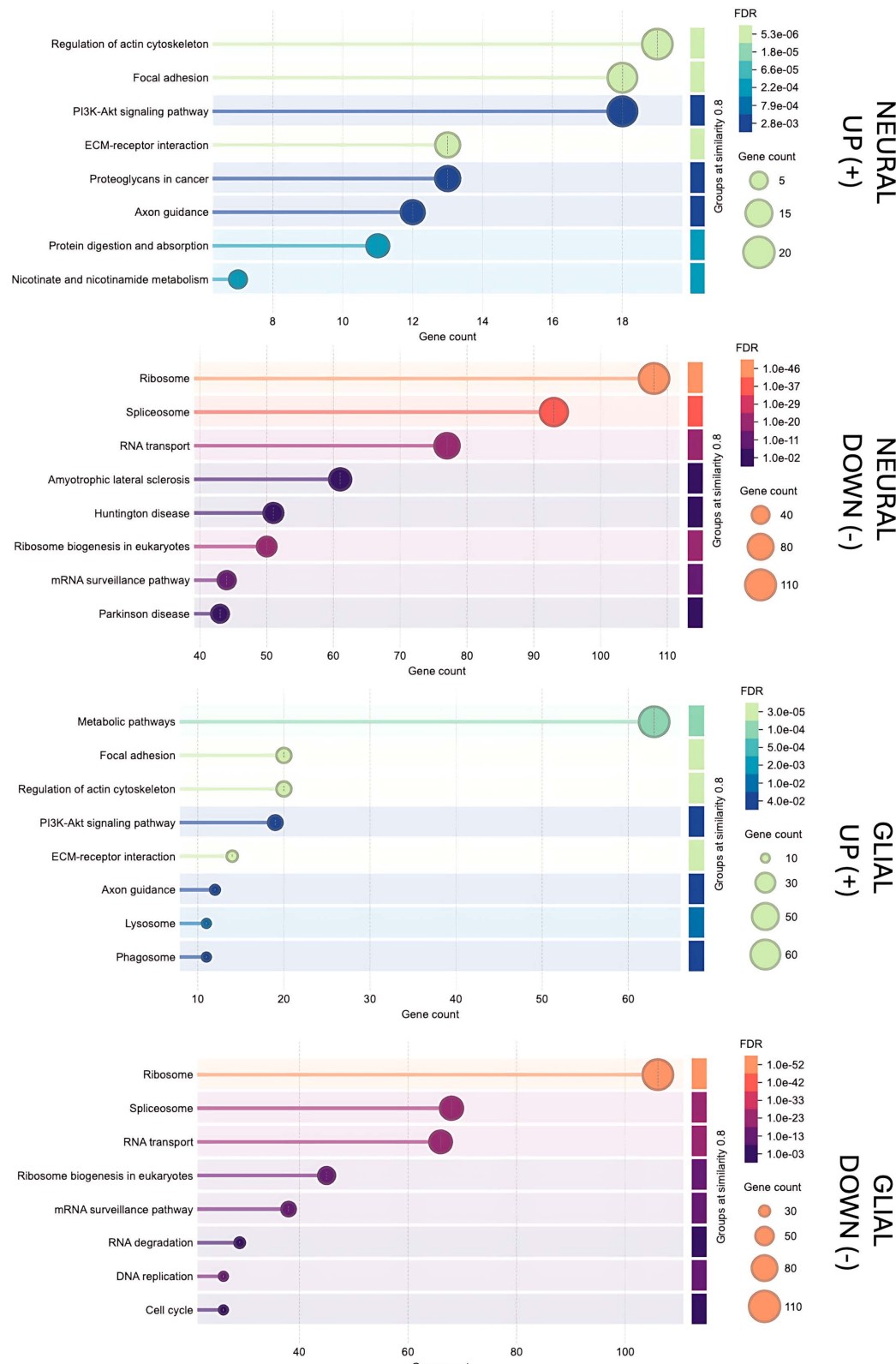

**Fig 7. KEGG pathway enrichment analysis of neural and glial MSCs compared to undifferentiated MSCs.** Differentially expressed proteins were analyzed using KEGG pathway enrichment analysis. From top to bottom: pathways upregulated in neural MSCs, pathways downregulated in neural

MSCs, pathways upregulated in glial MSCs and pathways downregulated in glial MSCs. Bubble size represents the number of proteins mapped to each pathway, and color indicates the false discovery rate (FDR). Pathways were considered significant at FDR<0.05. D1, D2, D3, D4, D5 and D7 was used in this experiment.

## Discussion

Our results demonstrate that bone marrow MSCs derived from MS patients exhibit morphological changes, reduced proliferation and migration capacity, and upregulation of neural and glial markers when cultured in neural induction medium supplemented with specific growth factors. Immunostaining confirmed the expression of the glial markers GFAP, SOX2, and PDGFRα, indicating a commitment toward glial-like phenotype, with SOX2 and PDGFRα being significantly more expressed in glial-MSCs compared with undifferentiated MSCs. Additionally, the detection of β-III-tubulin and NEFL supports neuronal differentiation potential, with NEFL being significantly more expressed in neural-MSCs compared to undifferentiated MSCs.

These findings were further supported by molecular characterization using mass spectrometry analysis. The proteomic heatmap revealed distinct clustering of MSC, G-MSC, and N-MSC populations, confirming phenotypic separation after the three-week transdifferentiating process. Notably, the expression patterns in the heatmap show generally consistent trends among the six donors, with no obvious donor-specific outliers. This reproducibility challenges prior reports suggesting that MSC donor heterogeneity impacts transdifferentiation efficiency [30]. The PCA analysis reveals that MSCs undergo dynamic changes over time during transdifferentiating process. After three weeks of transdifferentiation, the samples cluster distinctly from earlier time points along the principal components, indicating that the MSC transdifferentiating is a gradual, continuous process rather than a singular event. This focus on the time aspect sets our study apart from earlier work, which often looked at MSC transdifferentiation at a single point without capturing its ongoing nature. The robust downregulation of proliferation and MSC identity markers, together with upregulation of neural and glial proteins, provides strong evidence that MSCs reproducibly undergo early-stage lineage commitment under growth factor driven conditions. The actin cytoskeleton undergoes extensive remodeling during the differentiation process of MSCs. Neural and glial MSCs shows upregulation of proteins connected with regulation of actin cytoskeleton in the KEGG pathway analysis. Transdifferentiating MSCs also showed an upregulation of axon guidance proteins, that are involved in body axis formation, neuronal migration and axonal growth [31]. PI3K-Akt signaling pathway is an important pathway to maintain cell survival during neural differentiation and this was also upregulated in neural and glial MSCs [32]. Interestingly, glial MSCs exhibited reduced proliferative activity despite enrichment of metabolic pathways. This suggests that increased metabolic activity may not primarily support cell proliferation but instead reflect metabolic reprogramming associated with lineage specification, cellular maintenance, or functional maturation [33]. Several pathways related to protein synthesis, RNA processing and cell cycle regulation were downregulated in glial MSCs. This downregulation is consistent with reduced proliferative activity and a shift away from highly proliferative MSC state. Also the downregulated pathways in neural MSCs are predominantly involved in processes related to protein synthesis and RNA processing.

Our findings align with prior research demonstrating that BM-MSCs can adopt neural-like morphologies and express neural and glial markers under controlled in vitro conditions. For instance, Harris et al. [34] reported that BM-MSCs from both MS patients and healthy controls differentiated into neural progenitor-like cells (MSC-NPs) using a serum-free neural progenitor maintenance medium with EGF and bFGF over 21 days. They observed elevated expression of neural stem cell markers (e.g., nestin, neurofilament-M, GFAP) and the chemokine receptor CXCR4 after one week, along with reduced levels of MSC markers such as CD90 and SMA. These changes mirror our results, particularly the upregulation of neuro-glial markers and downregulation of mesenchymal identifiers. However, although their differentiation approach was similar to ours, their study did not include proteomic analysis, which limits the depth of molecular characterization compared with our work.

Consistent with previous findings, we observed reduced proliferation and migration rates in transdifferentiating cells compared to undifferentiated MSCs, indicative of a functional shift from mesenchymal to neural progenitor-like state [35]. Proteomic profiling further confirmed the downregulation of MSC identity proteins (e.g., CD166, CD105) and proliferation markers (e.g., CDK1, PCNA), supporting that MSCs shift toward a neural progenitor-like sate. While some earlier studies have relied primarily on immunostaining, our combination of morphology, functional assays, and proteomics provides more robust evidence for early-stage transdifferentiation.

The potential for MSCs to transdifferentiate into neural lineages has been debated for decades. Early reports suggested that MSCs could acquire neural-like morphologies and marker expression in vitro, however, many of these observations have been critiqued on methodological grounds. For example, commonly used neural markers can be expressed in undifferentiated MSCs or induced rapidly in ways inconsistent with true differentiation, and morphological changes may reflect stress responses rather than lineage commitment [30]. In contrast, our findings show increased expression of several neural and glial markers across different cellular functions, supported by protemic analysis and immunostaining. Importantly, these molecular changes were accompanied by reduced proliferation and migration rates, as assessed by independent functional assays. Together, the consistency across multiple methodologies and the shift away from a highly proliferative, migratory MSC phenotype argue against a generalized stress response and instead support a gradual transition toward neural- and glial-like progenitor states.

Moreover, reviews of MSC biology highlight that the broad therapeutic effects observed in experimental models are more possibly mediated by paracrine signaling and modulation of the host microenvironment than by transdifferentiation into neurons or glia [36]. Consistent with these challenges, our study does not include functional assays of neuronal activity. The changes we observed likely reflect progression toward progenitor-like neural states rather than full differentiation into mature neurons or oligodendrocytes.

However, a limitation of the study is the relatively small sample size of seven donors. An inclusion of a larger cohort would enhance the findings. Additionally, the lack of a StemPro NSC medium control without growth factors represents a limitation of this study. Without this condition, it cannot be fully excluded that the basal medium itself contributes to the induction of neural-associated features. This limitation may partially explain the similar morphology and overlapping protein expression observed between neural and glial MSCs. The observed reductions in proliferation and migration rate in the transdifferentiating MSCs may cause practical challenges for scaling up cell production and delivery in therapeutic contexts. Furthermore, part of the confluency in the migration assay is due to proliferation. However, a profound difference in migration can be observed: In the proliferation assay (Fig 3), we observe a ~ 25% difference in proliferation between MSCs and the transdifferentiating cells at 48 hours (43% vs 32% confluency), while in the migration assay (Fig 2), we observe a 35% difference already after 24 hours (78% vs 51% confluency).

These findings suggest that growth factor-mediated priming can effectively direct MSCs toward neuro-glial lineages, offering a promising strategy for regenerative therapies targeting the CNS. Partial transdifferentiation into glial- and neural-like phenotypes may enhance MSC compatibility with the CNS microenvironment, facilitating interactions with endogenous neurons, oligodendrocyte precursor cells, and glia to support neuroprotection and repair [34].

Moreover, transdifferentiating MSCs may acquire a more CNS-relevant supportive phenotype, reflected by increased expression of proteins associated with neural and glial markers. The protein Nestin a type VI intermediate filament, first described in neural stem cells, but are also expressed in several tissues and progenitor cells. Nestin is essential to stem cell functions like self-renewal, differentiation and migration and were more expressed in N-MSCs [37]. The elevated expression of NOTCH2 in glial MSCs is consistent with activation of CNS relevant progenitor signaling pathways, as Notch signaling remains active in the adult brain and contributes to the maintenance and regulation of neural and glial progenitor states [38]. The elevated expression of vimentin in glial-MSCs is consistent with a glial-like phenotype resembling immature or reactive astrocyte states, which are increasingly recognized as context-dependent and often supportive during CNS repair [39]. GAP43, a protein associated with axonal growth and neural structural plasticity, was most highly

expressed in neural-MSCs [40]. In addition, neural-MSCs showed elevated expression of NCAM1, a neural cell adhesion molecule that plays a central role in axon guidance, neurite outgrowth, and neuronal migration [41]. PDGFRα was most expressed in glial-MSCs, consistent with a phenotype related to glial progenitor phenotype, as PDGFRα is a key receptor involved in oligodendrocyte progenitor signaling [42]. Pre-conditioning MSCs toward a neural fate may also reduce the risk of unwanted mesodermal differentiation in vivo, thereby improving safety for clinical applications [34].

Preclinical studies using MS mouse models highlight the therapeutic potential of neural-primed MSCs. For instance, Brown et al. demonstrated that MSC-derived neural stem cells (NSCs) outperformed undifferentiated MSCs in alleviating symptoms of experimental autoimmune encephalomyelitis (EAE) diseased mice. NSCs promoted myelination, neuroprotection and neurogenesis and induced anti-inflammatory response [43].

These effects have been validated in additional EAE models, showing neural-differentiated MSCs lowered clinical scores, enhanced remyelination and reduced inflammation [44–46]. Together, this suggests that neural-differentiated MSCs may be a proper candidate for regenerative therapy.

Clinically, this approach has progressed to human trials. In a phase I/II study by Harris et al., intrathecal administration of MSC-NPs to 20 patients with progressive MS was safe, with 7 patients showing sustained improvements in EDSS scores over two years [47]. Analysis of cerebrospinal fluid (CSF) revealed immunomodulatory shifts, including reduced pro-inflammatory chemokine CCL2 and increased levels of IL-8, HGF, and CXCL12. However, a challenge is the reduced proliferation of transdifferentiating MSCs.

Although our study focused on BM-MSCs from MS patients, the observed capacity for glial-and neural-like cells has broader implications for regenerative approaches in other neurodegenerative diseases, such as Parkinson's disease and Alzheimer's disease, where neuronal and glial dysfunction are central pathological features.

This study establishes that bone marrow MSCs from MS patients can be reproducibly driven toward neuro-glial lineages over a 21-day transdifferentiating period using neural medium with EGF, bFGF, and PDGF-AA. Evidenced by distinct proteomic profiles separating undifferentiated MSCs and transdifferentiating MSCs. Future research should focus on larger, more diverse patient cohorts, in vivo validation in MS animal models, and strategies to address limitations in cell proliferation and migration.

## Supporting information

**S1 Fig. Migration rate of MSCs, G-MSCs and N-MSCs.**
(XLSX)

**S2 Fig. Proliferation rate of MSCs, G-MSCs and N-MSCs.**
(XLSM)

**S3 Fig. Immunostaining of MSCs, G-MSCs and N-MSCs.**
(XLSX)

**S1 File. Proteomic raw data.**
(XLSX)

**S2 File. Proteomic raw data.**
(XLSX)

**S3 File. Proteomic raw data.**
(XLSX)

**S4 File. Combined proteomics normalized.**
(XLSX)

## Author contributions

**Conceptualization:** Marie Ytterdal, Lars Bø, Christopher Elnan Kvistad, Torbjørn Kråkenes.

**Data curation:** Marie Ytterdal.

**Formal analysis:** Marie Ytterdal, Torbjørn Kråkenes.

**Methodology:** Marie Ytterdal, Casper Eugen Sandvik, Torbjørn Kråkenes.

**Resources:** Trygve Holmøy.

**Software:** Marie Ytterdal.

**Supervision:** Lars Bø, Christopher Elnan Kvistad, Torbjørn Kråkenes.

**Writing – original draft:** Marie Ytterdal, Casper Eugen Sandvik.

**Writing – review & editing:** Trygve Holmøy, Lars Bø, Christopher Elnan Kvistad, Torbjørn Kråkenes.

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
