## [Decision Letter · Decision Letter 0]

10 Nov 2025

Dear Dr. Ytterdal,

Thank you for submitting your manuscript to PLOS ONE. After careful consideration, we feel that it has merit but does not fully meet PLOS ONE’s publication criteria as it currently stands. Therefore, we invite you to submit a revised version of the manuscript that addresses the points raised during the review process.

We look forward to receiving your revised manuscript.

Kind regards,

Vikash Chandra, PhD

Academic Editor

PLOS ONE

Journal Requirements:

2. Please ensure you have included the registration number for the clinical trial referenced in the manuscript.

3. Please note that PLOS One has specific guidelines on code sharing for submissions in which author-generated code underpins the findings in the manuscript. In these cases, we expect all author-generated code to be made available without restrictions upon publication of the work. Please review our guidelines at https://journals.plos.org/plosone/s/materials-and-software-sharing#loc-sharing-code and ensure that your code is shared in a way that follows best practice and facilitates reproducibility and reuse.

“M.Y.: PhD stipend from Helse Vest, Bergen, Norway”

6. We note that your Data Availability Statement is currently as follows: “All relevant data are within the manuscript and its Supporting Information files. Patient information is limited to what is already included in the method section of the manuscript.”

Additional Editor Comments (if provided):

The manuscript "The neural transdifferentiation potential of bone marrow mesenchymal stem cells" is an interesting investigation in the field of stem cell therapeutics. But it needs significant improvement before acceptance. So, revise the manuscript in the light of reviewers' comments.

Reviewers' comments:

Reviewer's Responses to Questions

**Comments to the Author**

1. Is the manuscript technically sound, and do the data support the conclusions?

Reviewer #1: No

Reviewer #2: Yes

2. Has the statistical analysis been performed appropriately and rigorously?

Reviewer #1: No

Reviewer #2: No

3. Have the authors made all data underlying the findings in their manuscript fully available?

The PLOS Data policy requires authors to make all data underlying the findings described in their manuscript fully available without restriction, with rare exception (please refer to the Data Availability Statement in the manuscript PDF file). The data should be provided as part of the manuscript or its supporting information, or deposited to a public repository. For example, in addition to summary statistics, the data points behind means, medians and variance measures should be available. If there are restrictions on publicly sharing data—e.g. participant privacy or use of data from a third party—those must be specified. requires authors to make all data underlying the findings described in their manuscript fully available without restriction, with rare exception (please refer to the Data Availability Statement in the manuscript PDF file). The data should be provided as part of the manuscript or its supporting information, or deposited to a public repository. For example, in addition to summary statistics, the data points behind means, medians and variance measures should be available. If there are restrictions on publicly sharing data—e.g. participant privacy or use of data from a third party—those must be specified.

Reviewer #1: Yes

Reviewer #2: Yes

4. Is the manuscript presented in an intelligible fashion and written in standard English?

Reviewer #1: Yes

Reviewer #2: Yes

Reviewer #1: The study by Ytterdal et al entitled “The neural transdifferentiation potential of bone marrow mesenchymal stem cells” examines the ability of mesenchymal stem cells (MSCs) to transdifferentiate to a neuro and glial lineages. The study addresses a decades-long controversy regarding the ability of MSCs to transdifferentiate from mesodermal to neuroectodermal lineage, a phenomenon suggested to be due to phenotypic plasticity and experimental artifact rather than true lineage conversion. While the current study offers some important insights in terms of timing and proteomic changes during transdifferentiation, there remains many deficits in the study including a lack of quantitation and statistical analysis that limit the overall interpretation that acquisition of a neural/glial phenotype by MSCs represents transdifferentiation.

1) Introduction lines 77-82: The authors cite 3 clinical trials using MSCs in MS. The first 2 trials represent IV administration of MSCs in RRMS patients. The limitations of the 2 studies should be noted, including IV administration on a RRMS patient population that already benefit from highly effective therapies thus limiting the need for MSCs in this population. A description of Ref 17 should also be included in this paragraph because it represents IT administration and reported efficacy in a progressive patient population.

2) Include additional demographics on the six donors including MS subtype, EDSS, disease duration, gender, and age.

3) Figure 1 is subjective and could easily represent a biased interpretation.

a. The undifferentiated MSCs which were repeatedly passaged over the 3 weeks are shown at subconfluence whereas the differentiated MSCs are confluent. The morphology of representative confluent MSCs should be used to compare.

b. The advantage of using 6 separate MSC lines is to provide quantitative data. Any changes in morphology needs to be demonstrated for all 6 lines. The authors should apply high resolution imaging metrics using Image J or Cell Profiling to provide more robust and unbiased quantitation of basic morphological differences between MSCs, N- and G-MSCs from all 6 lines. At the very least, additional representative images of the other 5 lines should be included in the supplemental data. Figure legend should state that the images are representative.

c. The authors should address the morphological similarity between neural and glial differentiated cells despite claiming they represent different lineages. Other than being “different” from MSCs, please comment on how N or G MSC morphology compares to brain derived neural/glial cultures.

4) Figure 2 scratch assay:

a. Please justify the conclusion that the scratch assay is reflective of migratory capacity of MSCs. Clearly the proliferation rate of N- and G-MSCs is reduced and thus reflected in the reduced rate of closure in the scratch assay. The limitation of this assay in terms of assessing migratory capacity and its relationship to proliferation should be clearly delineated.

b. The discussion point line 454 “decreased migratory capacity could hinder homing to CNS lesions” should be removed or revised since migration in this context refers to specific chemotactic responses that are typically measured in a trans well assay.

c. The legend states average and SE from 3 different experiments. Please clarify whether the graph represents all 6 cell lines that were tested in 3 different experiments and how the average and SE were determined. Data from all 6 cells lines needs to be included along with statistical analysis to support any differences between MSCs and N/G-MSCs.

5) Figure 3 proliferation: See point 4c above regarding data representing all 6 cells lines and statistical analysis.

6) Figure 4 needs to include quantitative analysis of immunostaining in all 6 cell lines to demonstrate reproducibility. The figure showing a representative image of each marker is meaningless without quantitative analysis and statistics. There appears to be more GFAP staining in N/G MSCs but there are also more cells in the images. This reflects the fact that immunofluorescence is not ideal to quantitate protein expression and should be supported by a more quantitative method (Western blotting) in all 6 lines.

7) Figure 5, the PCA plot appears to represent 2 samples of MSCs, 2 samples of N and G MSCs at 7 days, and 3 samples of N and G MSCs at 14 and 21 days. Please clarify whether these are representative of only 1 cell line in duplicate/triplicate or 3 cell lines? Please clarify whether or not proteomics was performed in all 6 lines.

8) Figure 6

a. please clarify why there are 7-9 biological replicates when the methods states 6 separate cell lines and the PCA plot only shows 2-3.

b. What were the criteria for selecting the markers represented in the heatmap?

c. The proteomic data should be analyzed by performing unbiased enrichment analysis to identify pathways that are altered. Without pathway analysis, it’s not clear that the markers as shown were cherry picked or are meaningful representatives of cell proliferation or differentiation towards a certain lineage.

9) Figures 7, 8 and 9

a. The markers in these figures were from a predefined panel. Please explain the criteria for selection of the panel. Why are some of the markers not included in Figure 6?

b. Clarify, again, whether the averages in the graphs represent 1 or 3 or 6 separate cell lines (see comment to Figure 5).

c. If not representative of all 6 cell lines, then additional protein quantitation should be performed (i.e. western blot) to confirm these protein changes in all 6 lines.

d. Please perform statistical analysis on graphs in figures 7, 8 and 9.

10) Figure 9, the oligodendroglial markers chosen do not accurately represent oligodendrocyte lineages. PDGFA is not typically expressed in oligodendrocytes as it is the ligand for oligo marker and receptor PDGFRA. The authors should show more typical markers of oligodendrocytes such as olig1 and olig2.

11) Discussion

a. line 388 concludes that the expression patterns showed high consistency with minimal donor to donor variability. This statement is not supported by the data in all 6 cell lines as outlined in the above comments.

b. Line 433, when discussing a more CNS-relevant secretory profile, only one of the factors (PDGFA) is secreted the others are cytoplasmic or membrane associated. Please revise this paragraph to clarify how these specific proteins might be related to improved trophic support of differentiated MSCs.

12) A major question regarding transdifferentiation is whether or not it represents an in vitro artifact due to stress caused by altered culture conditions. StemPro MSC medium contains a growth supplement (serum alternative). How much of the results shown here in terms of proliferation, ‘migration’, and expression of proliferation markers are due to removal of the growth supplement when culturing in StemPro NSC media? Furthermore, is the N or G phenotype reversible, meaning that after 3 weeks if you put the cells back in StemPro MSC do they resume proliferation? An even more important question is whether or not mesodermal differentiation (adipocytes, osteocytes, chondrocytes) is lost after N or G differentiation. Without additional experiments to answer these questions, the discussion needs to include a more balanced and tempered interpretation of the current findings regarding transdifferentiation.

13) Address why G and N MSCs have similar morphology and protein expression despite exposure to different growth factors. The appropriate control would be StemPro NSC media without growth factors to show that the growth factors are indeed promoting differentiation into glial and neural lineages. Without this experiment, the possible alternative interpretations of this data should be included.

14) Include a more thorough discussion on the controversy around transdifferentiation of MSCs and the limitations of the current study. Multiple references from the 2000’s initiated the controversy underlying whether transdifferentiation occurs or is an in vitro artifact. Those references are omitted here but serve important perspective on the topic. Please discuss the limitations of the current study including the lack of functional data, lack of evidence of terminal differentiation, and lack of differentiation controls for proliferative effects of the culture media as outlined above. Furthermore, the overall lack of evidence in the literature supporting in vivo differentiation of MSCs or Neural MSCs into neural lineages upon transplantation is a major limitation to the interpretation that neural differentiation is relevant to the regenerative capacity of these cells.

Reviewer #2: The manuscript titled “The neural transdifferentiation potential of bone marrow mesenchymal stem cells” reports original research investigating the ability of MSCs from multiple sclerosis (MS) patients to acquire neural and glial-like characteristics in vitro. The study is well designed, technically sound, and supported by detailed proteomic analysis that enhances its scientific depth. The findings are consistent and reproducible, indicating early neuro-glial lineage commitment. Overall, the paper is clearly written and contributes meaningful molecular insight into MSC biology. However, several issues concerning methodological transparency, statistical rigor, and ethical reporting should be addressed before it can be considered for publication. Please find the detailed comments below.

Major Comments

1- The introduction does not sufficiently explain the scientific reasoning for selecting MSCs from MS patients rather than healthy donors. The rationale should clarify whether the intent was to model disease-specific MSC behavior, to evaluate the feasibility of autologous transplantation in MS, or to test whether the disease state influences differentiation potential. Without this context, the choice of donor source appears arbitrary and limits interpretation of the findings.

2- A section describing statistical methodology is missing. The results include qualitative claims of “significant” differences in migration and proliferation, but no information is provided on statistical tests, sample size, or p-values.

3- The fluorescence images are interpreted qualitatively. If possible, quantitative analysis (e.g., mean fluorescence intensity or proportion of marker-positive cells across replicates) should be included.

4- More detailed information about the six MS donors is needed, including sex distribution, MS subtype (progressive vs. relapsing), disease duration, and treatment history. These factors are relevant for assessing potential biological variability and for enabling reproducibility.

5- The manuscript lists “N/A” under Ethics Statement, although human-derived cells were used. Please revise to include the appropriate ethical approval (REK#159326) and confirm that written informed consent was obtained under the SMART-MS protocol.

6- The proteomic dataset is extensive, but the analysis is limited to marker-level observations. Incorporating pathway-level bioinformatic analyses (e.g., Gene Ontology or KEGG enrichment) would help identify broader biological processes and signaling pathways associated with transdifferentiation. This can be performed using the existing dataset and would considerably enrich the study’s interpretive depth.

7- Terminology: “Transdifferentiation”. Since functional neural or glial activity was not demonstrated, it would be more accurate to refer to the observed changes as lineage priming or phenotypic conversion rather than full transdifferentiation.

Minor Comments

1- Several typographical errors should be corrected (e.g., “trans-differentiated,” “neurodegeneratice”).

2- Confirm that data files in the Supporting Information fully correspond to the figures and text.

**Do you want your identity to be public for this peer review?** For information about this choice, including consent withdrawal, please see our  For information about this choice, including consent withdrawal, please see our  For information about this choice, including consent withdrawal, please see our  For information about this choice, including consent withdrawal, please see our Privacy Policy..

Reviewer #1: **Yes:**Violaine K. HarrisViolaine K. HarrisViolaine K. HarrisViolaine K. Harris

Reviewer #2: **Yes:**Niyaz Al-SharabiNiyaz Al-SharabiNiyaz Al-SharabiNiyaz Al-Sharabi

---

## [Author Response · Author response to Decision Letter 1]

16 Jan 2026

1. “Please ensure that your manuscript meets PLOS ONE's style requirements, including those for file naming. The PLOS ONE style templates can be found at https://journals.plos.org/plosone/s/file?id=wjVg/PLOSOne_formatting_sample_main_body.pdf and https://journals.plos.org/plosone/s/file?id=ba62/PLOSOne_formatting_sample_title_authors_affiliations.pdf”

This has been updated.

2. “Please ensure you have included the registration number for the clinical trial referenced in the manuscript.”

This is included in the manuscript.

3. “Please note that PLOS One has specific guidelines on code sharing for submissions in which author-generated code underpins the findings in the manuscript. In these cases, we expect all author-generated code to be made available without restrictions upon publication of the work. Please review our guidelines at https://journals.plos.org/plosone/s/materials-and-software-sharing#loc-sharing-code and ensure that your code is shared in a way that follows best practice and facilitates reproducibility and reuse.”

All code available.

4. “We note that the grant information you provided in the ‘Funding Information’ and ‘Financial Disclosure’ sections do not match.

When you resubmit, please ensure that you provide the correct grant numbers for the awards you received for your study in the ‘Funding Information’ section.”

This has been added.

5. “Thank you for stating the following financial disclosure:

“M.Y.: PhD stipend from Helse Vest, Bergen, Norway”

Please include this amended Role of Funder statement in your cover letter; we will change the online submission form on your behalf.”

This has been added to the cover letter.

6. “We note that your Data Availability Statement is currently as follows: “All relevant data are within the manuscript and its Supporting Information files. Patient information is limited to what is already included in the method section of the manuscript.

If there are ethical or legal restrictions on sharing a de-identified data set, please explain them in detail (e.g., data contain potentially sensitive information, data are owned by a third-party organization, etc.) and who has imposed them (e.g., an ethics committee). Please also provide contact information for a data access committee, ethics committee, or other institutional body to which data requests may be sent. If data are owned by a third party, please indicate how others may request data access.”

Yes all raw data are now included.

7. “If the reviewer comments include a recommendation to cite specific previously published works, please review and evaluate these publications to determine whether they are relevant and should be cited. There is no requirement to cite these works unless the editor has indicated otherwise.”

No extra citations were recommended.

Additional Editor Comments (if provided):

The manuscript "The neural transdifferentiation potential of bone marrow mesenchymal stem cells" is an interesting investigation in the field of stem cell therapeutics. But it needs significant improvement before acceptance. So, revise the manuscript in the light of reviewers' comments.

Reviewer #1: The study by Ytterdal et al entitled “The neural transdifferentiation potential of bone marrow mesenchymal stem cells” examines the ability of mesenchymal stem cells (MSCs) to transdifferentiate to a neuro and glial lineages. The study addresses a decades-long controversy regarding the ability of MSCs to transdifferentiate from mesodermal to neuroectodermal lineage, a phenomenon suggested to be due to phenotypic plasticity and experimental artifact rather than true lineage conversion. While the current study offers some important insights in terms of timing and proteomic changes during transdifferentiation, there remains many deficits in the study including a lack of quantitation and statistical analysis that limit the overall interpretation that acquisition of a neural/glial phenotype by MSCs represents transdifferentiation.

1) “Introduction lines 77-82: The authors cite 3 clinical trials using MSCs in MS. The first 2 trials represent IV administration of MSCs in RRMS patients. The limitations of the 2 studies should be noted, including IV administration on a RRMS patient population that already benefit from highly effective therapies thus limiting the need for MSCs in this population. A description of Ref 17 should also be included in this paragraph because it represents IT administration and reported efficacy in a progressive patient population.”

We thank the reviewer for their thorough evaluation of the manuscript and for the constructive comments. The introduction lines 77-98 has been revised to note the limitations of the 2 intravenous MSC trials conducted in RRMS patients. We have also included a description of Ref 17, emphasizing its intrathecal delivery approach and its relevance to progressive MS.

2) “Include additional demographics on the six donors including MS subtype, EDSS, disease duration, gender, and age.”

We agree with this suggestion. Additional donor demographic information has been included in Table 1 in the manuscript, including MS subtype, gender, age, disease duration (years since the diagnosis) and EDSS for all seven donors. We have also included which donor MSCs that have been used for each experiment in the figure legends in the Results section.

3) “Figure 1 is subjective and could easily represent a biased interpretation.

a. The undifferentiated MSCs which were repeatedly passaged over the 3 weeks are shown at subconfluence whereas the differentiated MSCs are confluent. The morphology of representative confluent MSCs should be used to compare.”

We agree with the reviewer and have now included morphology of confluent undifferentiated MSCs to be able to compare the morphology with the transdifferentiating MSCs.

b. “The advantage of using 6 separate MSC lines is to provide quantitative data. Any changes in morphology needs to be demonstrated for all 6 lines. The authors should apply high resolution imaging metrics using Image J or Cell Profiling to provide more robust and unbiased quantitation of basic morphological differences between MSCs, N- and G-MSCs from all 6 lines. At the very least, additional representative images of the other 5 lines should be included in the supplemental data. Figure legend should state that the images are representative.”

The images of cultured MSCs were included for illustrative purposes, to demonstrate typical cell appearance under the different culture conditions and were not intended to support quantitative morphological analyses. Quantitative high-resolution image analysis across all donors was beyond the scope of the current work, which focused on functional and molecular characterization rather than detailed morphological phenotyping. To avoid any potential misinterpretation, we have clarified in the Fig 1 legend that the images shown are representative and are provided for illustrative purposes only. We did not have camera in the cell lab when we started these experiments, so we don’t have images of all donor MSCs.

c. “The authors should address the morphological similarity between neural and glial differentiated cells despite claiming they represent different lineages. Other than being “different” from MSCs, please comment on how N or G MSC morphology compares to brain derived neural/glial cultures.”

We revised the first paragraph of the Results section to address the morphological similarities and differences between G-MSC and N-MSC, compared to brain-derived neural and glial cultures.

We agree that the N-MSC and G-MSC show overlapping morphological features in vitro, indicating they are not fully transdifferentiated into neural and glial lineages. In our cultures, both N-MSCs and G-MSCs display elongated and aligned growth patterns, which is less complex than those observed in primary brain-derived neural or glial cultures. Importantly, the N- and G-MSCs generated in this study represent progenitor-like populations rather than fully mature neurons or oligodendrocytes. We do not expect them to exhibit complex and highly specialized morphologies as terminally differentiated neurons or oligodendrocytes derived from brain tissue. Therefore, lineage distinction in our study is not concluded from morphology alone, but rather from lineage-associated marker expression and proteomics as presented in the manuscript.

4) Figure 2 scratch assay:

a. “Please justify the conclusion that the scratch assay is reflective of migratory capacity of MSCs. Clearly the proliferation rate of N- and G-MSCs is reduced and thus reflected in the reduced rate of closure in the scratch assay. The limitation of this assay in terms of assessing migratory capacity and its relationship to proliferation should be clearly delineated.”

A limitation of the migration assay is that there always will be a component of proliferation. However, if the reviewer would agree with our reasoning, there is a large difference in migration capacity as well: In the proliferation assay (Fig 3), we observe a ~25% difference in proliferation between MSCs and the transdifferentiating cells at 48 hours (43% vs 32% confluency). In the migration assay (Fig 2), we observe a 35% difference already after 24 hours (78% vs 51% confluency), which indicates that a profound part of the confluency is due to migration. We agree that this is not sufficiently explained in the text and have thus updated it in the discussion.

b. “The discussion point line 454 “decreased migratory capacity could hinder homing to CNS lesions” should be removed or revised since migration in this context refers to specific chemotactic responses that are typically measured in a trans well assay.”

We agree with the reviewer’s point. The statement suggesting that decreased migratory capacity could hinder homing to CNS lesions has been removed from the Discussion section to avoid overinterpretation beyond the scope of the migration assay used in this study.

c. “The legend states average and SE from 3 different experiments. Please clarify whether the graph represents all 6 cell lines that were tested in 3 different experiments and how the average and SE were determined. Data from all 6 cells lines needs to be included along with statistical analysis to support any differences between MSCs and N/G-MSCs.”

The data shown in the graph represent results obtained from three donor MSCs, generated from three independent experiments. Average values were calculated using data pooled from all three donor MSCs across the independent experiments. Statistical analyses comparing MSC with N- and G- MSCs were performed using the full dataset, as described in the revised Statistical Analysis section of the Methods. We have revised the legend in Fig 2, Methods and Results sections to clearly describe the experimental design, number of donor MSCs, and statistical approach used.

5) “Figure 3 proliferation: See point 4c above regarding data representing all 6 cells lines and statistical analysis.”

Three MSC donors were used in the proliferation assay and three independent experiments. We have clarified this and included statistical analysis and data representing three donor MSCs in the Methods, Results and Fig 3 legend in the manuscript.

6) “Figure 4 needs to include quantitative analysis of immunostaining in all 6 cell lines to demonstrate reproducibility. The figure showing a representative image of each marker is meaningless without quantitative analysis and statistics. There appears to be more GFAP staining in N/G MSCs but there are also more cells in the images. This reflects the fact that immunofluorescence is not ideal to quantitate protein expression and should be supported by a more quantitative method (Western blotting) in all 6 lines.”

We have now included a quantitative analysis of the immunostaining shown in Fig 4. Quantification was performed from two independent transdifferentiation experiments using MSCs from two different donors.

To measure the fluorescence intensity, 5 randomly selected fields (1000 µm x 1000 µm) for each coverslip with cells were analyzed in the software Qupath. This resulted in 10 measurements per marker and condition.

To account for differences in cell density between conditions, fluorescence data were normalized to undifferentiated MSCs, which were set to 100%, and the graph in Fig 4 therefore represent relative percentages of marker-positive cells normalized to MSC controls. Statistical comparisons between MSCs and N-MSCs, as well as MSCs and G-MSCs, were performed using Welch’s t-test. Mean values with 95% confidence intervals are now shown, and statistically significant differences or non-significant are indicated in the graphs. This has now been revised in the Methods and Results sections.

7) “Figure 5, the PCA plot appears to represent 2 samples of MSCs, 2 samples of N and G MSCs at 7 days, and 3 samples of N and G MSCs at 14 and 21 days. Please clarify whether these are representative of only 1 cell line in duplicate/triplicate or 3 cell lines? Please clarify whether or not proteomics was performed in all 6 lines.”

Proteomics were performed on MSCs derived from six donors. However, only one donor had proteomic data available at all three transdifferentiation time points (7, 14, and 21 days). To allow a direct comparison of proteomic changes across transdifferentiation stages, the PCA plot in Fig 5 was therefore generated using samples from this single donor only.

Due to instrument capacity limitations (a maximum of 18 samples per mass spectrometry run), samples at day 7 were analyzed in duplicate, while samples at days 14 and 21 were analyzed in triplicate, enabling inclusion of all time points within a single run. This has now been clarified in the Results and Fig 5 legend.

8) Figure 6

a. “please clarify why there are 7-9 biological replicates when the methods states 6 separate cell lines and the PCA plot only shows 2-3.”

The heatmap in Fig 6 includes data from six MSC donors. One donor is represented twice, as this donor was analyzed in two separate experimental batches performed a year apart, using the same transdifferentiation culture condition. These two samples are therefore shown separately in the heatmap. This clarification has now been added to the Methods section and to the Fig 6 legend.

b. “What were the criteria for selecting the markers represented in the heatmap?”

The markers included in the heatmap were selected based on a literature review of proteins commonly used to characterize MSCs, neural and glial lineage-associated phenotypes, and cell proliferation or progenitor states. Selection criteria included, established relevance to MSC identity or neural/glial differentiation and consistent detection across the proteomic da

---

## [Decision Letter · Decision Letter 1]

8 Feb 2026

Thank you for submitting your manuscript to PLOS ONE. After careful consideration, we feel that it has merit but does not fully meet PLOS ONE’s publication criteria as it currently stands. Therefore, we invite you to submit a revised version of the manuscript that addresses the points raised during the review process.

Please submit your revised manuscript by Mar 25 2026 11:59PM. If you will need more time than this to complete your revisions, please reply to this message or contact the journal office at plosone@plos.org. . . . A letter that responds to each point raised by the academic editor and reviewer(s). You should upload this letter as a separate file labeled 'Response to Reviewers'.A marked-up copy of your manuscript that highlights changes made to the original version. You should upload this as a separate file labeled 'Revised Manuscript with Track Changes'.An unmarked version of your revised paper without tracked changes. You should upload this as a separate file labeled 'Manuscript'.

We look forward to receiving your revised manuscript.

Kind regards,

Vikash Chandra, PhD

Academic Editor

PLOS One

Journal Requirements:

Additional Editor Comments:

Dear Authors,

Although most of the reviewers’ comments have been adequately addressed, the following concerns still require attention:

1. The newly added graphs in Figure 4 are very small and difficult to read. Please revise the figure to improve clarity and readability.

2. Please comment on whether SOX2, PDGF, and NEFL were also observed to be upregulated in the proteomic screen.

3. Line 448: The sentence beginning with “After 3 weeks of …” is incomplete, as the word “transdifferentiation” is missing.

Please revise the manuscript in light of the above comments and resubmit your revised manuscript for further consideration.

Reviewers' comments:

Reviewer's Responses to Questions

**Comments to the Author**

Reviewer #1: All comments have been addressed

Reviewer #2: All comments have been addressed

2. Is the manuscript technically sound, and do the data support the conclusions?

Reviewer #1: Yes

Reviewer #2: Yes

3. Has the statistical analysis been performed appropriately and rigorously?

Reviewer #1: Yes

Reviewer #2: Yes

4. Have the authors made all data underlying the findings in their manuscript fully available?

Reviewer #1: Yes

Reviewer #2: Yes

5. Is the manuscript presented in an intelligible fashion and written in standard English?

Reviewer #1: Yes

Reviewer #2: Yes

Reviewer #1: The authors have adequately addressed my comments and the manuscript is acceptable for publication. A few suggested minor edits that do not require another round of review.

1) The new graphs in Figure 4 are very small and difficult to read.

2) Suggestion that the authors comment on whether SOX2, PDGF, and NEFL were also observed to be upregulated in the proteomic screen.

3) Line 448: After 3 weeks of .... (transdifferentiation is missing)

Reviewer #2: Thanks to the authors for providing the revised version of the manuscript entitled “The neural transdifferentiation potential of bone marrow mesenchymal stem cells.” The authors have addressed all major and minor comments carefully and thoroughly. The rationale for using MSCs from MS patients is now clear, the statistical framework has been strengthened, and the quantitative analysis of imaging and proteomic data has improved the rigor of the study. The added donor information and corrected ethics statement further enhance transparency and reproducibility. Clarification of terminology and inclusion of pathway-level analyses have also brought the interpretation into closer alignment with the data.

The revised manuscript now presents a coherent and well-supported study.

---

## [Author Response · Author response to Decision Letter 2]

26 Mar 2026

Journal Requirements:

1. «If the reviewer comments include a recommendation to cite specific previously published works, please review and evaluate these publications to determine whether they are relevant and should be cited. There is no requirement to cite these works unless the editor has indicated otherwise. »

There was no recommendation to cite specific published work.

We have reviewed the reference list, no retracted articles were identified among the cited references, and therefore no changes to the reference list were required.

Additional Editor Comments:

1. «The newly added graphs in Figure 4 are very small and difficult to read. Please revise the figure to improve clarity and readability. »

Thank you for this important comment. The newly added graphs in Figure 4 have now been revised to improve clarity and readability.

2. «Please comment on whether SOX2, PDGF, and NEFL were also observed to be upregulated in the proteomic screen. »

Among these markers PDGFA was upregulated in the proteomic dataset. For inclusion in the proteomic heatmap, proteins were required to be consistently detected across all samples from all mass spectrometry batches. NEFL was detected only in one batch (where it was upregulated in neural-MSCs) and was therefore excluded from further analysis. For SOX2, the identifications did not meet the predefined confidence thresholds required for reliable protein assignment and quantitative analysis.

Discrepancies between the mass spectrometry data and the immunostaining likely arise from methodological differences. Antibody-based immunostaining allows sensitive and specific detection of predefined targets, whereas mass spectrometry–based proteomics relies on the presence, abundance, and analytical detectability of protein-derived peptides.

3. «Line 448: The sentence beginning with “After 3 weeks of …” is incomplete, as the word “transdifferentiation” is missing. »

The word transdifferentiation is now included.

---

## [Editor Report · Decision Letter 2]

29 Mar 2026

The neural transdifferentiation potential of bone marrow mesenchymal stem cells

PONE-D-25-55478R2

Dear Dr. Ytterdal,

We’re pleased to inform you that your manuscript has been judged scientifically suitable for publication and will be formally accepted for publication once it meets all outstanding technical requirements.

Kind regards,

Vikash Chandra, PhD

Academic Editor

PLOS One

---

## [Editor Report · Acceptance letter]

PONE-D-25-55478R2

PLOS One

Dear Dr. Ytterdal,

I'm pleased to inform you that your manuscript has been deemed suitable for publication in PLOS One. Congratulations! Your manuscript is now being handed over to our production team.

Kind regards,

on behalf of

Dr. Vikash Chandra

Academic Editor

PLOS One